2D versus 3D real time ultrasound with live xPlane imaging to visualize aortic and ductal arches: comparison between methods

Dell’Oro Stefania
Verderio Maria
Incerti Maddalena
http://orcid.org/0000-0002-1522-8732 Mastrolia Salvatore Andrea mastroliasa@gmail.com
Cozzolino Sabrina
Vergani Patrizia
Department of Maternal Fetal Medicine, Fondazione MBBM, San Gerardo Hospital, University of Milano-Bicocca , Monza , Italy
Erez Offer
Electronic publication date: 2018 Apr 6
Publication date: 2018
Volume: 6
Electronic Location ID: e4561
Received 2018 Feb 13; Accepted 2018 Mar 10
Copyright: © 2018 Dell’Oro et al.
Copyright year: 2018
Copyright holder: Dell’Oro et al.
License: This is an open access article distributed under the terms of the Creative Commons Attribution License, which permits unrestricted use, distribution, reproduction and adaptation in any medium and for any purpose provided that it is properly attributed. For attribution, the original author(s), title, publication source (PeerJ) and either DOI or URL of the article must be cited.
License URL: https://creativecommons.org/licenses/by/4.0/

Keywords: Real-time three-dimensional echocardiography, Congenital heart disease, Matrix probe, Second trimester screening, Prenatal diagnosis, Conotruncal anomalies

Funding: The authors received no funding for this work.

==============================
Background

The diagnosis of congenital heart defects is challenging, especially for what concerns conotruncal anomalies. Indeed, although the screening techniques of fetal cardiac anomalies have greatly improved, the detection rate of conotruncal anomalies still remains low due to the fact that they are associated with a normal four-chamber view. Therefore, the study aimed to compare real-time three-dimensional echocardiography with live xPlane imaging with two-dimensional (2D) traditional imaging in visualizing ductal and aortic arches during routine echocardiography of the second trimester of gestation.

Methods

This was an observational prospective study including 114 women with uncomplicated, singleton pregnancies. All sonographic studies were performed by two different operators, of them 60 by a first level operator, while 54 by a second level operator. A subanalysis was run in order to evaluate the feasibility and the time needed for the two procedures according to fetal spine position and operator’s experience.

Results

The measurements with 2D ultrasound were performed in all 114 echocardiographies, while live xPlane imaging was feasible in the 78% of the cases, and this was mainly due to fetal position. The time lapse needed to visualize aortic and ductal arches was significantly lower when using 2D ultrasound compared to live xPlane imaging (29.56 ± 28.5 s vs. 42.5 ± 38.1 s, P = 0.006 for aortic arch; 22.14 ± 17.8 s vs. 37.1 ± 33.8 s, P = 0.001 for ductal arch), also when performing a subanalysis according to operators’ experience (P < 0.05 for all comparisons). Feasibility of live xPlane proved to be correlated with the position of the fetal spine and the operator’s experience.

Discussion

To find a reproducible and standardized method to detect fetal heart defects may bring a great benefit for both patients and operators. In this scenario live xPlane imaging is a novel method to visualize ductal and aortic arches. We found that the position of the fetal spine may affect the feasibility of the method since, when the fetal back is anterior or transverse, the visualization of the correct view of three-vessels and trachea in order to set the reference line properly becomes more challenging. In addition, the fetal spine position influences the duration of the ultrasound examination. Regarding operator’s skills and experience, in our study a first level operator was able to perform the complete 2D and xPlane examination in a lower number of cases compared to second level operators. In addition, the time required for the complete examination was higher for first level operators. This means that this technique is based on an adequate operators’ expertise.

Introduction

Congenital heart defects (CHDs) are the most common form of congenital anatomical anomaly (Rocha et al., 2013).

They are associated with high morbidity and mortality rates, and their incidence is estimated to be 4–13 per 1,000 live births (Galindo et al., 2009; Sivanandam et al., 2006).

Antenatal screening offers a number of advantages such as establishing a strategy for peripartum management and screening for co-existing abnormalities, eventually allowing intrauterine intervention in some cases (Franklin et al., 2002).

However, the diagnosis of CHDs is challenging, especially for what concerns conotruncal anomalies. Indeed, although the screening techniques of fetal cardiac anomalies have greatly improved, the detection rate of conotruncal anomalies still remains low due to the fact that they are associated with a normal four-chamber view (Paladini et al., 1996; Sivanandam et al., 2006; Tometzki et al., 1999). At the beginning, the study of the fetal heart was based only on four-chamber view, while today, international and national (Italian Society of Obstetric and Gynecological Ultrasound, 2006, 2010, 2015) guidelines (Carvalho et al., 2013) consider left and right ventricular outflow tracts (LVOT and RVOT, respectively) as an important part of fetal cardiac screening examination, while three-vessels (3V), and three-vessels and trachea (3VT) are not mandatory due to the technically challenging acquisition and interpretation (International Society of Ultrasound in Obstetrics & Gynecology, 2006; Carvalho et al., 2013), although strongly recommended in order to increase and improve operators’ confidence with these scans.

The importance of improving the detection rate of CHDs, with special attention to conotruncal anomalies assessing the 3VT, as well as the aortic and ductal arches, in addition to the four-chamber view, LVOT and RVOT, was suggested by several authors (Allan, 2004; Espinoza et al., 2007b).

However, performing all these sonographic scans needs systematic training and may depend on the operators’ skills and experience (World Health Organization, 1998).

According to the above mentioned literature, there is a need to find a method that could improve the detection rate for CHDs, and especially of conotruncal anomalies, taking into consideration the possibility of a modification in the spatial relationships among cardiac chambers and great vessels throughout gestation (Espinoza et al., 2007a). This geometric three-dimensional (3D) change can be even more prominent in case of cardiac anomalies (Yuan et al., 2011), although the study of cardiac anomalies is beyond the scope of the present study.

Live xPlane imaging, a new kind of real-time 3D echocardiography, using a matrix-array probe, enabling the visualization of the pulsating fetal heart in real time, akin to real-time gray-scale scanning (Acar et al., 2005; Sugeng et al., 2003; Xiong et al., 2009, 2012a, 2012b, 2013a, 2013b; Yuan et al., 2011), might be useful to the purpose of improving the detection rate of CHDs according to geometric 3D changes of the fetal heart. In fact, this modality allows the simultaneous visualization of real-time high-resolution view of two planes, oriented in different directions, giving a satisfactory spatial resolution (Taddei et al., 2007).

Therefore, the objective of the present study was to evaluate to feasibility of real-time 3D echocardiography with live xPlane imaging compared to two-dimensional (2D) traditional imaging in visualizing ductal and aortic arches, analyzing the potential impact of factors influencing the feasibility of the technique, such as position of the fetal spine and operator’s experience and skills.

Materials and Methods

Setting and eligibility criteria

This was a prospective observational study including all women with singleton pregnancies from 18 + 1 to 21 + 6 weeks of gestation, attending routine echocardiography of the second trimester, between March 2015 and December 2016 at our Department of Maternal Fetal Medicine (Fondazione MBBM, San Gerardo Hospital, University of Milano-Bicocca, Monza, Italy). Women with indications for fetal echocardiography (chromosomal abnormalities, abnormal first trimester screening, fetal malformations, hereditary maternal diseases associated with cardiac defects, maternal infections, maternal disorders such as pregestational diabetes, phenylketonuria, autoimmune diseases, use of drugs with potential teratogenicity), were excluded from the study.

Medical and obstetrical history was investigated before enrollment in order to find any exclusion criteria. One hundred and fourteen women were enrolled in the study after meeting the inclusion criteria and accepting to participate. Informed consent was obtained in all cases.

Data collection

All ultrasound examinations were performed by using iU22 ultrasound scanner (Philips Ultrasound; Philips Medical System, Bothell, WA, USA), with two different probes: for the 2D ultrasound, the C5-1 probe was used, while the 3D scan was performed with the matrix X6-1 transducer.

The ultrasound scans were executed alternatively by two sonographers: a first level operator and a second level operator. Operators were defined according to the Report of WHO Study Group on training in diagnostic ultrasound. Therefore, the first level operator was defined as able to (a) perform common examinations safely and accurately; (b) recognize and differentiate normal anatomy and pathology; (c) diagnose common abnormalities within certain organ systems; (d) recognize when referral for a second opinion is indicated. The second level operator was defined as a specialist in Fetal Medicine able to (a) accept and manage referrals from Level 1 practitioners; (b) recognize and correctly diagnose almost all pathology within the relevant organ systems; (c) perform basic, non-complex ultrasound-guided invasive procedures; (d) teach ultrasound to trainees and to Level 1 practitioners; (e) conduct some research in ultrasound.

The first step of sonographic evaluation was the description of the position of the fetal spine.

Then, all women underwent 2D ultrasound and, starting from the four-chamber view, the time to visualize ductal and aortic arches in longitudinal section was measured.

The criteria for a successful imaging of the aortic arch view were that the whole course of the ascending aorta, transverse aortic arch, aortic isthmus, and upper part of descending aorta were visualized in continuity (Fig. 1). The criteria for the successful imaging of the ductal arch were the visualization of continuity of the right ventricle, pulmonary valve, pulmonary artery, ductus arteriosus, and upper part of the descending aorta, together with the short axis of ascending aorta (Fig. 2).

Figure 1 Aortic arch view, using live xPlane imaging.

On the left, the 3VT view, on the right, the aortic arch view, obtained by this new method.

Figure 2 Ductal arch view, using live xPlane imaging.

On the left, 3VT view, on the right, the ductal arch view, obtained by this new method.

The detailed method of visualization of the aortic arch and the ductal arch view with live xPlane imaging was defined as follows. A successful demonstration of the arches using live xPlane imaging required the correct orientation of the great vessels in the 3VT view. For this reason, the first step was to obtain the four-chamber view, then slightly tilting the transducer toward the fetal head in order to visualize the 3VT view. After that, the two planes were displayed in real time by the activation of live xPlane imaging function. The original 2D image was visualized on the left side of the screen (primary plane). The transducer was then moved in a way that either the pulmonary artery or the aorta was lying parallel to the direction of the ultrasound beam. A secondary plane across the reference line was displayed in the right window by moving the reference line on the primary plane. In our study, the 3VT plane was displayed on the left side. The reference line was adjusted in the primary plane to be placed along the center of the pulmonary trunk and descending aorta in the 3VT view, allowing the ductal arch view to be displayed in the right window. By the slight adjustment of the position of the transducer, moving the reference line through the center of the transverse aortic arch and descending aorta in the 3VT views, the aortic arch was visualized within the right window.

In our study, depending on fetal position, we displayed the aortic arch view at first and then the ductal arch view.

The exam was defined not doable when the two arches were not visualized within 5 min.

The study has been performed in accordance with the ethical standards laid down in the 1964 Declaration of Helsinki and its later amendments (Helsinki Declaration 1975, revision 2013) and approved by the Institutional Review Board Committee of San Gerardo Hospital (Monza, Italy) (IRB No. 3022015).

Statistical analysis was performed with IBM SPSS Statistic version 21 (Released 2012, Version 21.0. IBM SPSS Statistics for Windows, IBM, Armonk, NY, USA). Data on continuous variables with normal distribution were presented as mean ± SD, and compared between study groups using Student’s t-test. Continuous variables not normally distributed and ordinal variables were presented as median with inter-quartile range (IQ range), and statistical analysis was performed using Mann–Whitney test. Categorical data were shown in numbers and percentages, while their differences were assessed by Chi-square. Fisher’s exact test was used when appropriate. A P-value < 0.05 was considered as statistically significant.

Results

A total of 114 pregnant women of different ethnic groups were enrolled in this study. The mean maternal age was 33.07 ± 5.23 (19–45) years with a mean body mass index of 24.82 ± 4.17 (17.74–39.3). The mean gestational age at sonographic examination was 20.3 ± 0.5 weeks (18.6–21.5).

In six cases (5.3%) the fetus was in transverse lie, while in 108 exams (94.7%) fetuses were in longitudinal lie, of them 62 cases in vertex presentation (57.4%) and 46 in breech presentation (42.6%).

The visualization rate of the aortic arch and ductal arch with 2D ultrasound was performed in all 114 cases (100%), while the examination was completed in 89 cases (78.1%) using live xPlane imaging, of them 92 with aortic arch examination (80.7%) and 93 ending with ductal arch examination (81.5%) (Table 1).

Table 1 Feasibility with 2D ultrasound and live xPlane.

	Feasibility 2D (n = 114)	Feasibility live xPlane (n = 114)	P value	
Four chamber-aortic arch	114 (100%)	92 (80.7%)	<0.001	
Four chamber-ductal arch	114 (100%)	93 (81.5%)	<0.001	
Note:

Data are presented as number (percentage).

Table 2 shows a comparison between the time averages required with the two different methods. The timing with 2D ultrasound was significantly lower compared to live xPlane imaging starting from the four-chamber view until visualization of either aortic (29.56 ± 28.5 s vs. 42.5 ± 38.1 s, P = 0.006) as well as ductal arch (22.14 ± 17.8 s vs. 37.1 ± 33.8 s, P = 0.001).

Table 2 Average of times with 2D ultrasound and live xPlane.

	2D (n = 114)	Live xPlane (n = 89)	P value	
Four chamber-aortic arch (s)	29.56 ± 28.5	42.5 ± 38.1	0.006	
Four chamber-ductal arch (s)	22.14 ± 17.8	37.1 ± 33.8	0.001	
Note:

Data is presented as mean ± SD.

Compared to 2D ultrasound, live xPlane was more doable when the fetal back was in the right (100% vs. 100%), posterior (100% vs. 100%), or in left (100% vs. 93.8%) longitudinal position, while the feasibility was very low when the fetal back was anterior (100% vs. 24%) or transverse (100% vs. 33.3%) (Table 3).

Table 3 Feasibility of 2D ultrasound vs. live xPlane according to fetal spine position.

Fetal position	2D (n = 114)	Live xPlane (n = 114)	P value	
Anterior	26/26 (100)	6/25 (24)	<0.001	
Posterior	22/22 (100)	24/24 (100)	1	
Right	28/28 (100)	27/27 (100)	1	
Left	32/32 (100)	30/32 (93.8)	0.856	
Inferior	6/6 (100)	2/6 (33.3)	<0.001	
Superior	0/0 (0)	0/0 (0)	n/a	
Total	114/114 (100)	89/114 (78.07)	<0.001	
Note:

Data is presented as number (percentage).

In addition to feasibility of both techniques, a comparison of the time averages to perform 2D ultrasound vs. xPlane was done according to fetal spine position. Compared to live xPlane, 2D ultrasound was faster when the fetal back was in anterior (55.65 ± 34.14 s vs. 139.5 ± 62.84 s, P = 0.002), or in right longitudinal position (42.78 ± 28.65 s vs. 73.29 ± 50.64 s, P = 0.023), while it was slower when the back was in inferior position (80.33 ± 45.85 s vs. 46.0 ± 12.72 s, P = 0.042). When the spine was posterior (44.72 ± 25.33 s vs. 48.87 ± 51.58 s, P = 0.256) or in left position (55.75 ± 41.99 s vs. 64.0 ± 38.1 s, P = 0.377), times to achieve the scans were similar (Table 4).

Table 4 Average times need for 2D ultrasound vs. live xPlane according to fetal spine position.

Fetal position/time (sec)	2D (n = 114)	Live xPlane (n = 89)	P value	
Anterior	55.65 ± 34.14	139.5 ± 62.84	0.002	
Posterior	44.72 ± 25.33	48.87 ± 51.58	0.256	
Right	42.78 ± 28.65	73.29 ± 50.64	0.023	
Left	55.75 ± 41.99	64.0 ± 38.1	0.377	
Inferior	80.33 ± 45.85	46.0 ± 12.72	0.042	
Superior	0	0	n/a	
Total	51.71 ± 35.36	67.4 ± 51.29	0.111	
Note:

Data is presented as number (percentage).

Echocardiographies were performed by two different operators: 60 (52.6%) of them were performed by a first level operator, while 54 (47.4%) were executed by a second level operator.

The first level operator, had a successful visualization rate of the aortic and ductal arch with 2D ultrasound in all 60 cases (100%), while the complete live xPlane imaging was performed only in 38 cases (63.3%), of them 40 with aortic arch examination (66.6%) and 41 ending with ductal arch examination (68.3%). Regarding the second level operator, the visualization of the aortic and ductal arch with 2D ultrasound was performed in all 54 cases, while it was possible to perform aortic arch examination in 52 cases (96.29%) and ductal arch examination in another 52 women (96.29%), having the complete live xPlane imaging in 51 cases (94.4%) (Table 5).

Table 5 Feasibilty of 2D ultrasound vs. live xPlane according to operator level.

First level operator	2D ultrasound	Live xPlane	P value	
Four chamber-aortic arch	60/60 (100)	40/60 (66.6)	<0.001	
Four chamber-ductal arch	60/60 (100)	41/60 (68.3)	<0.001	
Second level operator	2D ultrasound	Live xPlane	P value	
Four chamber-aortic arch	54/54 (100)	52/54 (96.29)	0.56	
Four chamber-ductal arch	54/54 (100)	52/54 (96.29)	0.56	
Note:

Data is presented as number (percentage).

The comparison between the time averages required with the two different methods, according to operator’s level is shown in Table 6. A higher amount of time to perform the examination with live xPlane compared to 2D ultrasound was required for both first level as well as second level operator. For the first level operator, the time with 2D ultrasound was significantly lower compared to live xPlane imaging starting from the four-chamber view until visualization of either aortic (30.25 ± 29.25 s vs. 51.62 ± 36.89 s, P = 0.002) as well as ductal arch (25.51 ± 20.62 s vs. 39.09 ± 34.23 s, P = 0.03). Also for second level operator, the time needed to visualize the aortic arch using 2D ultrasound was significantly lower (28.79 ± 28.17 s vs. 35.5 ± 37.9 s, P = 0.02) as was for ductal arch (18.38 ± 13.21 s vs. 35.5 ± 33.8 s, P = 0.001), compared to live xPlane.

Table 6 Average of times with 2D ultrasound vs. live xPlane according to operator level.

First level operator	2D ultrasound	Live xPlane	P value	
Four chamber-aortic arch (s)	30.25 ± 29.25	51.62 ± 36.89	0.002	
Four chamber-ductal arch (s)	25.51 ± 20.62	39.09 ± 34.23	0.03	
Second level operator	2D ultrasound	Live xPlane	P value	
Four chamber-aortic arch (s)	28.79 ± 28.17	35.5 ± 37.9	0.02	
Four chamber-ductal arch (s)	18.38 ± 13.21	35.5 ± 33.8	0.001	
Note:

Data is presented as mean ± SD

Figure 3 shows the distribution of the times with 2D vs. 3D live xPlane, related with the spine position and divided according to operator’s level. The median for the 2D method was always lower or similar to 3D live xPlane, for both the first and second level operator. This was true except when the fetal back was posterior since, in this case, the second level operator performed live xPlane in a lower time compared to 2D ultrasound.

Figure 3 Distribution of the times with 2D vs. 3D live xPlane, related with the spine position and divided according to operator’s level.

Discussion

Principal findings of the study: (1) Live xPlane imaging allows a better identification of the aortic and ductal arches without the risk to confound them, while this is not true for the 2D method where the longitudinal visualization of the arches may mislead the operator due to their similar characteristics; (2) this technique can be useful in fetal echocardiography during the second trimester executed by an experienced operator; and (3) the time required to complete live xPlane examination compared to 2D ultrasound examination of the fetal heart is dependent on both fetal position and operator experience.

The need for additional techniques in the screening for CHDs: Although screening of CHDs has developed greatly in the last years, according to new guidelines, the detection rate of fetal conotruncal anomalies is still low (Galindo et al., 2009; Paladini et al., 1996; Sivanandam et al., 2006; Tometzki et al., 1999), especially among non-experienced operator (Tegnander & Eik-Nes, 2006). This is mainly because these anomalies usually are associated with a normal four-chamber view. Detection rate can only be increased by including additional views of the outflow tracts, as recommended both for SIEOG and ISUOG guidelines (Carvalho et al., 2013; Italian Society of Obstetric and Gynecological Ultrasound, 2015). This would further include left and right outflow tract view and 3VT. However, realizing these four views is not easy and needs systematic training and depends on the operators’ experience (Tegnander & Eik-Nes, 2006). Therefore, to find a simple, reproducible, and standardized method to detect fetal heart defects may bring a great benefit for both patients and operators, especially non-experienced ones.

Three-vessels and trachea views can be obtained as easily as the four-chamber view, which may be useful as a complementary view in routine screening for CHD (Vinals, Heredia & Giuliano, 2003; Yagel et al., 2002). However, according to the available literature, which proposes the use of 3VT in routine screening, we decided to study the aortic and ductal arches with live xPlane, since easily obtainable from the 3VT, despite the study of fetal heart does not include the study of aortic and ductal arches. In fact, aortic and ductal arch are a useful tool to screen for conotruncal anomalies (Espinoza et al., 2007b).

Some researchers proposed to use spatiotemporal image correlation, a form of reconstructed 3D echocardiography, to improve the detection rate of fetal conotruncal anomalies (DeVore et al., 2003; Espinoza et al., 2007b; Goncalves et al., 2006; Yagel et al., 2002). However, this method is influenced by fetal breathing and movements and is subjected to movement artifacts (Goncalves et al., 2003).

Conversely, live xPlane, based on real-time 3D processing, allows simultaneous display of two planes in real time and, for this reason, is not associated to artifacts.

Live xPlane imaging and its utility according to fetal cardiac geometry: Live xPlane imaging is described as a novel and relatively simple method, with a good sensibility and specificity, simple to teach and learn (Xiong et al., 2012a). The application of a matrix probe in scanning fetal heart had been reported by several researchers (Goncalves et al., 2003, 2006), but there are only few studies providing a detailed description of its methodology in the visualization of the recommended sections of the fetal heart (Xiong et al., 2009, 2012a, 2012b, 2013a, 2013b; Yuan et al., 2011).

In the available literature, live xPlane has been employed to visualize some specific cardiac structures, in order to improve the detection rate of CDHs. For example, Xiong et al. (2009, 2013a) evaluated the feasibility of this method for the visualization of interventricular septum, while Yuan et al. (2011) used live xPlane to examine simultaneously the four-chamber, the LVOT and the angle between these structures. Furthermore, Xiong et al. (2012b) reported the methodology of acquiring and examining the screening planes, with the advantages and disadvantages of this method.

Yuan et al. (2011), use xPlane to study the rotation angle from four-chamber view to LVOT and investigate factors affecting the angles: they describe how abnormal rotation angles and angle span may be associated with congenital heart disease with conotruncal disorders.

Instead, Espinoza et al. describe how the fetal cardiac geometry occurs during the gestation, particularly before 26 weeks of gestation. These observations indicate that the spatial relationships between cardiac chambers and great vessels are not constant throughout gestation (Espinoza et al., 2007a).

What is the time needed for the evaluation of the aortic and ductal arches? Live xPlane is strongly related with the spine position and the fetal movements. In comparison with 2D ultrasound, it requires a longer time average because, in order to get the aortic and ductal arches, the reference line should be placed in the center of the pulmonary trunk and in the aorta. There is no literature analyzing the time average needed to obtain these scans, as we did in the present study.

What is the effect of fetal spine position in the evaluation of the aortic and ductal arches? Feasibility of live xPlane proved to be strongly correlated with position of the fetal spine. Literature suggests live xPlane to be feasible in 100% of cases, but the available studies were performed when the back was posterior only (Xiong et al., 2013b). No studies correlated the feasibility of the technique with the fetal spine position. Indeed, when the back is anterior or transverse, the feasibility is very low, because of the difficulty to visualize the correct view of 3VT in a way to apply the method putting correctly the reference line, as we observed in our study. Of interest, as described within our study, the fetal spine position influences the duration of the ultrasound examination.

Strengths and limitations of the study: The main advantage of live xPlane imaging is that it is relatively simple and that there are no movement artifacts because live xPlane imaging is performed in real time. Major limitation of this study is that the reference line only comes from the midline of the transducer and cannot be manipulated from all directions. In fact, the reference line could only be placed in parallel position concerning the ultrasound beam. This is the reason why, when the fetal back is anterior or transverse, it is really difficult to obtain the visualization of the arches. Thus, if the fetus is in an unfavorable position, the acquisition time will be much longer.

Conclusion

Live xPlane it is a relatively simple method and may potentially be a useful tool in the ultrasound evaluation for the diagnosis of fetal conotruncal anomalies allowing a visualization of the aortic and ductal arches and improving the detection rate of CHDs. This is especially true for contruncal anomalies, taking into consideration the possibility of a modification in the spatial relationships among cardiac chambers and great vessels throughout gestation (Espinoza et al., 2007a) and the fact that this geometric 3D change can be even more prominent in case of cardiac anomalies (Yuan et al., 2011), although the study of cardiac anomalies is beyond the scope of the present study. Live xPlane is dependent on the position of the fetal spine and needs an adequately trained operator in order to be performed. Increasing the operators’ expertise is crucial in order to improve the detection of cardiac anomalies using this innovative technique.

Supplemental Information

Supplemental Information 1 Data file.

Click here for additional data file.

Additional Information and Declarations

Competing Interests

Author Contributions

Human Ethics

Data Availability

Dr. Salvatore Andrea Mastrolia is an Academic Editor for PeerJ.

Stefania Dell’Oro analyzed the data, contributed reagents/materials/analysis tools, prepared figures and/or tables, authored or reviewed drafts of the paper, approved the final draft.

Maria Verderio conceived and designed the experiments, performed the experiments, authored or reviewed drafts of the paper, approved the final draft.

Maddalena Incerti performed the experiments, prepared figures and/or tables, authored or reviewed drafts of the paper, approved the final draft.

Salvatore Andrea Mastrolia conceived and designed the experiments, analyzed the data, contributed reagents/materials/analysis tools, prepared figures and/or tables, authored or reviewed drafts of the paper, approved the final draft.

Sabrina Cozzolino performed the experiments, prepared figures and/or tables, authored or reviewed drafts of the paper, approved the final draft.

Patrizia Vergani conceived and designed the experiments, authored or reviewed drafts of the paper, approved the final draft.

The following information was supplied relating to ethical approvals (i.e., approving body and any reference numbers):

The study has been performed in accordance with the ethical standards laid down in the 1964 Declaration of Helsinki and its later amendments (Helsinki Declaration 1975, revision 2013) and approved by the Institutional Review Board Committee of San Gerardo Hospital (Monza, Italy).

The following information was supplied regarding data availability:

The raw data are provided in Supplemental Dataset Files.

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
