# Peer review of "D versus 3D real time ultrasound with live xPlane imaging to visualize aortic and ductal arches: comparison between methods"

_PeerJ, doi:10.7717/peerj.4561_

## Round 0.1 · original submission · Major Revisions

· Academic Editor

Major Revisions

Dear Dr. Mastrolia,

Your paper has been reviewed by two experts in the field who raised major criticisms. One of them suggested to reject the manuscript, however, I think that the topic is interesting and timely and this manuscript has the potential to be of use to the community, therefore, I would like to reconsider your manuscript for publication that will address all the concerns raised by the reviewers.

·

Basic reporting

This is a prospective observational study evaluating the feasibility of using 3D Real Time Ultrasound with live xPlane imaging for visualization of the aortic and ductal arches during fetal echocardiography and comparing this technique to the standard 2D approach.

Main issues:
1- Manuscript needs professional English editing
2- Avoid using the term cardiac malformations thoughtout the text. “cardiac anomalies” would be a better fit.

Experimental design

1- Feasibility of 3D real time ultrasound with live xPlane imaging for visualization of aortic and ductal arches was established by Xiong et al. This article mainly added the analysis of various factors (fetal position, movements, operator experience) affecting successful acquisition of arches views and the difference in time needed between 2D and Xplane imaging.
This needs to be stressed in the objectives of the study.

“Xiong Y, Chen M, Chan LW, Ting YH, Fung TY, Leung TY, and Lau TKA 2012aA A novel way of
visualizing the ductal and aortic arches by real-time three-dimensional ultrasound with live xPlane
imaging. Ultrasound Obstet Gynecol 39:316-321A 10A1002/uogA9081”


2- The authors mentioned that the study was conducted between March 2015 and December 2016 and included all pregnant women at time of fetal anatomic survey with no indication for fetal echo. My question is how was these 114 women selected?
During almost 22 months only 114 women scanned? Please explain in methodology section.

3- Discussion section of the abstract is a repeat of what was already mentioned in the introduction. This section should focus on value of your results.

Validity of the findings

The findings are useful in understanding the limitations and various factors affecting (fetal position, movements, operator experience) affecting successful acquisition of arches views and the difference in time needed between 2D and Xplane imaging.

Reviewer 2 ·

Basic reporting

Comments on the paper “2D versus 3D Real Time Ultrasound with live xPlane imaging to visualize aortic and ductal arches: comparison between methods” by Stefania Dell'Oro et al.
The authors evaluated 114 patients between 18+1 to 21+6 weeks of gestation using normal 2D ultrasound and a commercially available technique based on live 3D ultrasound (XPlane)
• No significant differences were observed, significantly longer period of time to obtain the required planes with XPlane .

Experimental design

• No abnormal cases were included.
• The potential benefits of the new technique cannot be assessed in this study

Validity of the findings

• No real benefit with the new technique can be achieved

• The potential benefits of the new technique cannot be assessed in this study

Additional comments

• No significant differences were observed, significantly longer period of time to obtain the required planes with XPlane .
• No real benefit with the new technique can be achieved
• No abnormal cases were included.
• The potential benefits of the new technique cannot be assessed in this study
• The results showed that in normal pregnancies for operators level 1 and 2 2D ultrasound is still the standard for cardiac evaluation

---

## Round 0.2 · accepted · Accept

· Academic Editor

Accept

The authors has addressed the comments of the reviewer and improved the manuscript substantially and it can now accepted for publication